Multiomics analysis reveals flavonoid accumulation and biosynthesis across different cultivation years and localities of Gongronemopsis tenacissima (Dai-Bai-Jie)

Wang Mengqi 1
Gu Yunxia 1
Shan Liming 1
Li Chunyu 1
Yuan Ertai 1
Li Ge lige19800221@163.com 2
Liu Xiaoli kmxunzi@aliyun.com 1
1 College of Chinese Material Medica, Yunnan University of Chinese Medicine , Kunming , Yunnan , China
2 Institute of Medicinal Plant Development, Chinese Academy of Medical Sciences & Peking Union Medical College , Jinghong , Yunnan , China
Gelfand Mikhail
Electronic publication date: 2025 Dec 16
Publication date: 2025
Volume: 13
Electronic Location ID: e20439
Received 2025 Feb 5; Accepted 2025 Oct 31
Copyright: ©2025 Wang et al.
Copyright year: 2025
Copyright holder: Wang et al.
License: This is an open access article distributed under the terms of the Creative Commons Attribution License, which permits unrestricted use, distribution, reproduction and adaptation in any medium and for any purpose provided that it is properly attributed. For attribution, the original author(s), title, publication source (PeerJ) and either DOI or URL of the article must be cited.
License URL: https://creativecommons.org/licenses/by/4.0/

Keywords: Metabolome, Transcriptome, Rhizosphere microbes, Flavonoids, Gongronemopsis tenacissima

Funding: Xishuangbanna Prefecture Science and Technology Plan Project No. 202401001 This study was financially supported by the Xishuangbanna Prefecture Science and Technology Plan Project (No. 202401001). The funders had no role in study design, data collection and analysis, decision to publish, or preparation of the manuscript.

==============================
Background

The dried root of Gongronemopsis tenacissima (Roxb.) S. Reuss, Liede & Meve, traditionally known as Dai-Bai-Jie, is a medicinal plant used by the Dai ethnic group, primarily for detoxification. Due to extensive use, wild resources have become increasingly scarce, prompting domestication efforts in China. However, the accumulation patterns of secondary metabolites—particularly flavonoids, the main detoxifying components—and their biosynthetic mechanisms remain unclear.

Methods

This study investigated differences in flavonoid accumulation and transcriptional regulation in Dai-Bai-Jie cultivated for one, two, and three years at high altitudes, and three years at low altitudes. Transcriptome and widely targeted metabolome analyses were conducted. A total of 1,495 metabolites were identified using ultra-performance liquid chromatography coupled with tandem mass spectrometry (UPLC-MS/MS), with 943 showing differential accumulation across the four groups. All flavonoids were classified into six clusters using k-means clustering. Flavonoid levels were generally higher in plants cultivated for two and three years, with total metabolite content also more abundant in these groups. Two-year cultivation was recommended as the optimal harvesting strategy.

Results

A regulatory relationship was observed between genes such as phenylalanine ammonia-lyase, CYP73A, 4-coumarate: coenzyme A ligase, and flavonol synthase, and the flavonoid components in Dai-Bai-Jie. No significant differences were found in Shannon, Chao1, or abundance coverage estimator (ACE) indices of rhizosphere microorganisms across different cultivation years and locations.

Conclusions

This study elucidates the mechanisms of flavonoid accumulation and supports the scientific rationale for optimal harvesting years of Dai-Bai-Jie. The findings provide a valuable foundation for guiding large-scale cultivation and reducing reliance on wild resources.

Introduction

Gongronemopsis tenacissima (Roxb.) Moon S.Reuss, Liede & Meve is a traditional medicine utilized by the Dai ethnic group, commonly known as Dai-Bai-Jie, and holds significant value in the ethnomedical traditions of Southeast Asia. In the Dai language, it is referred to as “Ya Jie Xian Da”, symbolizing its ability to purge the body of numerous toxins. This medicinal herb has long been utilized in Dai-inhabited regions such as Xishuangbanna, Dehong, Ximeng, Menglian, Xinping, Yuanjiang, Mojiang, and Puer in China, as well as neighbouring countries like Laos and Myanmar (Li, Michael & Douglas, 1995). The root of G. tenacissima is employed in folkloric medicine named Dai-Bai-Jie for detoxification purposes. It is recognized for its efficacy in counteracting toxicities resulting from various sources, including food, animals, and environmental factors such as heat, water, and fire burns. Additionally, it is used to relieve throat discomfort and swelling caused by excessive heat toxicity. With a rich historical background in traditional medicine, Dai-Bai-Jie has been incorporated into contemporary hospital preparations at institutions like the Xishuangbanna Dai Hospital. These formulations include Bai-jie Capsules, Ya-jie Gahan, and Banna Coolant. Modern pharmacological research has shown that “Dai-Bai-Jie” possesses inhibitory effects on cancer cells, protects against liver damage caused by certain drugs, demonstrates anti-HIV activity, possesses antioxidant properties, and exhibits antibacterial activities (Gao et al., 2014; Li et al., 2021).

Currently, various bioactive compounds have been isolated from Dai-Bai-Jie, including organic acids, polyoxyprogesterone glycosides, volatile oils, and pyrrole alkaloids (Liao et al., 2016; Pang et al., 2018; Song et al., 2018; Song et al., 2021). These discoveries not only deepen our understanding of the medicinal properties of this herb but also open up potential avenues fortherapeutic applications in modern medicine.

For a long time, Dai-Bai-Jie was incorrectly identified as the dried root of Dregea sinensis Hemsl., which belongs to the genus Dregea of the Asclepiadaceae family (Lin, Zhuan & Zhao, 2003). However, a pivotal study conducted in 2014 revealed that Dai-Bai-Jie is, in fact, the dried root of Marsdenia tenacissima, a species within the genus Marsdenia (Li et al., 2014; Li et al., 2023). This identification was based on comprehensive molecular and morphological analyses utilizing DNA fragments such as psbD-trnT, trnL F, and ITS, along with observations of leaf morphology and floral characteristics. It is important to note that the “tong-guan-teng” mentioned in the Chinese Pharmacopoeia, known for its broad-spectrum anticancer properties is associated with G. Cavalieri (formerly known under M. Cavalieri (Chen et al., 2022; Li et al., 2014). Current scientific investigations have revealed significant differences in the chemical composition and therapeutic effects of these two species. Specifically, Dai-Bai-Jie is primarily indicated for antidotal properties and management of gastrointestinal disease, while anticancer activity is chiefly attributed to G. cavalieri. In 2022, M. tenacissima was reclassified into the genus Gongronemopsis and is now referred to as Gongronemopsis tenacissima (Roxb.) (Liede-Schumann et al., 2022).

Flavonoids are secondary metabolites found widely in plants, possessing a variety of functions including antioxidant, anti-inflammatory, antitumor, antiviral, antibacterial, anti-vascular sclerosis, and anti-liver fibrosis activities (Fang et al., 2023; Wang et al., 2020; Zhang et al., 2023a; Zhang et al., 2023b). Recent studies suggested that their protective effect on intestinal mucosal barrier function may play a role in detoxification mechanisms (Yang et al., 2020). According to Dai medical theory, the occurrence of disease is closely linked to imbalances among the four cosmic elements within the body, which can be triggered by the presence of toxins (Zhang et al., 2023a; Zhang et al., 2023b). Such imbalances may stem from disturbances in antioxidant defences and disparities between pro- and anti-inflammatory factors. Notably, recent studies have demonstrated a correlation between the levels of total flavonoids and total polyphenols in Dai-Bai-Jie with its antioxidant and anti-inflammatory activities (Zhang et al., 2023a; Zhang et al., 2023b). Therefore, flavonoids may represent the most significant active component for detoxification properties of Dai-Bai-Jie.

Due to the extensive utilization of Dai-Bai-Jie, wild resources are becoming increasingly scarce. Fortunately, significant advancements have been made in the artificial cultivation technology for G. tenacissima, leading to small-scale cultivation in Xishuangbanna, Yunnan. Under natural conditions, the harvesting period for the roots of G. tenacissima is typically determined by empirical knowledge and generally occurs after at least two years of growth. Similarly, under cultivation conditions, the harvest period is usually 2–3 years, primarily considering the biomass of the roots.

Despite these advancements, the accumulation patterns of flavonoids in Dai-Bai-Jie under varying cultivation conditions remain unclear. To address this knowledge gap, this study investigated the flavonoid accumulation patterns and influencing factors of Dai-Bai-Jie from a multi-omics perspective, which may lead to a better understanding of the metabolic accumulation mechanism of Dai-Bai-Jie and facilitate the scientific determination of optimal harvesting years for this medicinal plant.

Materials and Methods

Plant materials and sampling

The roots of one-year-old (CR1), two-year-old (CR2), and three-year-old (CR3) cultivated G. tenacissima (Dai-Bai-Jie) were collected from Menghun County, Xishuangbanna Dai Autonomous Prefecture, Yunnan, China (E 100.38°, N 21.82°; 1,179 m) in November 2022 (Fig. 1). Additionally, the roots of three-year-old Dai-Bai-Jie (CR4) cultivated in South Medicine Garden (E 100.79°, N 22.00°; 533.57 m) also located in Xishuangbanna Dai Autonomous Prefecture, Yunnan Province, China, were gathered. Each plant was divided into two sections: one for transcriptome sequencing and the other for metabolome analysis, with three biological replicates per sample. Furthermore, the rhizosphere soil (CM1, CM2, CM3, CM4) corresponding to each plant (CR1, CR2, CR3, CR4) was collected and utilized for 16S rRNA and ITS analysis.

Figure 1 The sample used in this study.

(A) Total plant of Dai-Bai-Jie. (B) Root of cross-sections at different planting years. CR1: farmed for one year, CR2: farmed for two years, CR3, farmed for three years.

Metabolite extraction and UPLC-MS/MS analysis

After the freeze-dried samples were crushed (30 Hz, 1.5 min), the extraction solution (70% methanol water pre-cooled to −20 °C) was added, and the mixture was vortexed for 30 s. Subsequently, the samples were vortexed six times (once every 30 min) and centrifuged at 12,000 rpm for 3 min. The supernatant was then filtered through a microporous filter membrane with a pore size of 0.22 µm and stored in an injection vial for Ultra Performance Liquid Chromatography (UPLC-MS/MS) analysis.

Ultra High Performance Liquid Chromatography (ExionLC™ AD) was employed for sample collection and analysis, utilizing an Agilent SB-C18 column (1.8 µm, 2.1 mm × 100 mm). The mobile phase A consisted of 0.1% formic acid in water, while the mobile phase B was acetonitrile containing 0.1% formic acid. The column temperature was maintained at 40 °C, and the automatic sampler temperature was set to 4 °C. The flow rate was adjusted to 0.35 mL/min, and the injection volume was 2 μL.

Applied Biosystems 6500 QTRAP was used for analysis. The typical ion source parameters were as follows: electrospray ionization (ESI) temperature of 500 °C; ion spray voltage (IS) of 5,500 V in positive ion mode and −4,500 V in negative ion mode; ion source gas I (GSI), gas II (GSII), and curtain gas (CUR) were set to 50, 60, and 25 psi, respectively. The collision-induced dissociation parameters were set to high. SCIEX Analyst workstation software (version 1.6.3) was used for Multiple Reaction Monitoring (MRM) data collection and processing.

Data processing, metabolite identification, and statistical analysis

Using MS-Converter, MS raw data files were converted into TXT format for further analysis. An internal R program, along with a specialized database, was employed for peak detection and annotation. In metabolite identification, the parameters matched include Q1 accurate molecular weight, secondary fragments, retention time, and isotopes. The qualitative and quantitative mass spectrometric analysis of metabolites in project samples is based on the MetWare Database (MWDB) and MRM.

The identification of metabolites in the project samples is based on the precise mass of the metabolites, MS2 fragments, the isotopic distribution of MS2 fragments, and retention time (RT). By employing the intelligent secondary spectrum matching method developed in-house, the secondary spectrum and RT of the metabolites in the project samples are intelligently matched one by one with the secondary spectrum and RT in the company database. The MS tolerance and MS2 tolerance are set to 20 ppm, while the RT tolerance is set to 0.2 min. The level of substance identification is categorized into three levels: Level 1, Level 2, and Level 3, with decreasing accuracy in that order. Level 1  represents the highest accuracy with MS/MS spectrum and RT match score ≥ 0.7, MS/MS spectrum and RT match score between 0.5 and 0.7 of level 2. The level 3 indicates that the sample substance’s Q1, Q3, RT, DP, and CE match consistently with the database substance. The raw metabonomic data can be found at Zenodo (https://doi.org/10.5281/zenodo.17222367).

Prior to analysis, the raw data underwent preprocessing to filter out low-quality ion signals. After obtaining the organized data, SIMCA (version 16.0.2) software was used for Principal Component Analysis(PCA)and Orthogonal Partial Least Squares Discriminant Analysis (OPLS-DA), which were used to explore the metabolic patterns and identify differential metabolites (DAMs) with p-values < 0.05 and VIP (variable importance in projection) > 1.

RNA-seq processing and data analysis

Total RNA was extracted and purified from the above samples. The extracted RNA was tested for purity, concentration, and integrity. After the samples were qualified, the mRNA was isolated and purified by Oligo (dt) for the construction of the cDNA library. Illumina Novaseq 6000 sequencing was performed after the library was qualified. Fastp software (Chen et al., 2018) was used for quality control on the raw data.

After obtaining clean reads, Trinity assembly software is used to splice the clean reads to obtain reference sequences for subsequent analysis, trinity assembly software was used to stitch the clean reads to obtain reference sequences for subsequent analysis.

The RSeQC software (Wang, Wang & Li, 2012) was used to evaluate the quality of transcriptome dataand to analyze the sequencing data after passing the quality evaluation. Fragments per Kilobase Million (FPKM) (Trapnell et al., 2010) was used to estimate gene expression level. The transcriptome assembly was assessed in terms of their completeness and the percentage of complete, fragmented, and missing fragments by using the BUSCO 5.3.2 (https://busco.ezlab.org, Simão et al., 2015). DESeq2 (Love, Wolfgang & Simon, 2014; Varet et al., 2016) was used for differential expression analysis between samples. The corrected p-value and FDR (False Discovery Rate) were used as the key indicators for the screening of differentially expressed genes (DEGs). Weighted gene co-expression network analysis (WGCNA) was used to find the gene modules that are co-expressed and construct the hierarchical clustering tree. The statistical power of this experimental design, calculated in RNASeqPower is 0.70.

The whole transcript data set can be found in the National Center for Biotechnology Information (NCBI) database (BioProject ID: PRJNA996325).

Reverse transcription-quantitative polymerase chain reaction validation

We selected five genes associated with flavonoid synthesis for reverse transcription-quantitative polymerase chain reaction (RT-qPCR) according to FPKM value (Forkmannm & Martens, 2001; Zou et al., 2016). GAPDH was used as a reference gene and all genes used in this study are listed in Table 1. cDNA was synthesized using MonScript™ RTIII All-in-One Mix with ds DNase (Monad). According to the instructions of QuantiNova SYBR Green PCR Kit (Qiangen), RT-qPCR was performed. The total volume of the system was 10 μL, including five μL 2x SYBR Green PCR Master Mix, 0.7 μL upstream primer with 0.7 µM, 0.7 μL downstream primer with 0.7 µM, one µL cDNA with ≤100 ng/reaction, 2.55 µL RNase-free water, 0.05 µL QN ROX Reference Dye.

Table 1 Primer of Five genes.

Gene	Sequence (5′-3′)	Product size	
Cluster-43408.2
FLS	F: TGATGAATGGGAAGCCCGAG
R: TAGCGGTCCTGTTTTGGCTT	175 bp	
Cluster-46899.5
FLS	F: AGCCCTTGAAGAATTTGGTTGT
R: ATCTCTTGTAAAGGCCGATCAAA	114 bp	
Cluster-51734.2
CYP73A	F:GGACCTGGCTAAGGAAGTGT
R: TGTGAAGAAAGGCACCGTCA	166 bp	
Cluster-60047.2
4CL	F: GCATCCGTGGCGATCAAATC
R: TGCCACTTGGAACCCTTTGT	179 bp	
Cluster-63886.1
PAL	F: CATGCCCTCCTCAACAACGA
R: GGACCTGCACTCCTTGATCC	171 bp	
GAPDH	F: GGCATTGTCGAGGGTCTCAT
R: CCGGTGCTGCTGGGAATAAT	131 bp	

Microbial DNA extraction, 16S rRNA, and ITS gene sequencing

Genomic DNA was extracted using CTAB (Nobleryder). A total of 30 μL PCR amplification system was as follows: Phusion® High-Fidelity PCR Master and high fidelity polymerase Mix (New England Biolabs) 15 μL, Primer 1 μL, DNA 5–10 ng, ddH2O. 16S V4 regional primer 515F (5′-GTGCCAGCMGCGCGGGGGTAA-3′) and 806R (5′-GGACTACHVGGGGTWTCTAAT-3′) were used to identify bacterial diversity. ITS5-1737F (5′-GGAAGTAAAAGTCGTAACAAGG-3′) and ITS2-2043R (5′- GCTGCGTTCTTCATCGATGC-3′) were used to identify fungal diversity. Reaction procedure was set at 98 °C for 1 min, followed by 40 cycles at 98 °C for 10 s, 0 °C for 38 s, and 72 °C for 30 s, 72 °C extension for 5 min finally. PCR products were sequenced on the NovaSeq6000 platform (Maiwei Biotechnology Company).

Results

RNA-seq analysis and DEGs identification

We performed high-throughput transcriptome sequencing on the CR1, CR2, CR3, and CR4 of Dai-Bai-Jie, with three biological replicates per sample. In total, we obtained 78.27 GB of clean data. The clean data of all samples were not less than 6 GB of clean data. The percentages of bases with a Q30 quality score were greater than 90% for all samples. After assembling and splicing, 85,346 unigenes were obtained. A BUSCO analysis was performed to evaluate the completeness, recovering 253 out of 255 conserved eukaryotic genes (99.2%) (Fig. 2A).

Figure 2 The DEGs in the four groups were analyzed by KEGG metabolic pathway.

(A) CR1-CR2. (B) CR1-CR3. (C) CR2-CR3. (D) CR3-CR4.

Using the criteria of |log2Fold Change|≥ 1 and FDR <  0.05, we screened for DEGs. The results revealed that 15,255, 8,170, 10,529, and 8,225 DEGs were identified in the comparisons of CR1 vs. CR2, CR1 vs. CR3, CR2 vs. CR3, and CR3 vs. CR4, respectively. Among these, 654 common DEGs were shared across CR1, CR2, CR3, and CR4. Specifically, there were 6,043 unique DEGs identified in the comparison of CR1 vs. CR2, 1,243 unique DEGs in CR1 vs. CR3, 2,720 unique DEGs in CR2 vs. CR3, and 2,957 unique DEGs in CR3 vs. CR4 (Fig. 2B).

The DEGs in the four groups were analyzed using the Kyoto Encyclopedia of Genes and Genomes (KEGG) metabolic pathway. The results showed that the DEGs of CR1 vs. CR2, CR1 vs. CR3, CR2 vs. CR3, and CR3 vs. CR4 were annotated to 144, 140,143, and 140 KEGG metabolic and biosynthetic pathways, respectively. Notably, the “Metabolic pathways” category emerged as the most frequently annotated, encompassing 2,492, 1,428, 1,669, and 1,432 genes in each comparison, respectively. This was closely followed by the “biosynthesis of secondary metabolites” category, which annotated 1,375, 800, 930, and 808 genes, respectively. The “Plant-pathogen interaction” pathway was annotated to 514, 271, 364, and 401 genes (Fig. 3).

Figure 3 Transcriptome analysis results.

(A) BUSCO completeness assessments of the Dai-Bai-Jie transcriptome. (B) WGCNA clustering tree. (C) Venn Diagram representing the number of DEGs among four group sample.

WGCNA displayed that DEGs are divided into 27 co-expression modules of CR1, CR2, CR3, and CR4. Among them, the turquoise module has the highest number of genes with 11,313, followed by the blue module with 5550 genes, and the least is the white module, which has 101 genes (Fig. 2C).

RT-qPCR validation

The RT-qPCR results for the five targeted genes indicated that four of them (excluding cluster-60047.2) displayed a general consistency with the relative transcript abundance observed in the transcriptome analysis. This concordance validates the reliability of the RNA-seq data (Fig. 4).

Figure 4 RNA-seq analysis of Dai-Bai-Jie and the qRT-PCR validation of five genes.

RNA-seq analysis of Dai-Bai-Jie and the qRT-PCR validation of five genes.

Metabolomic profiling

A total of 1495 metabolites were identified from Dai-Bai-Jie using UPLC-MS/MS. These included 378 amino acids and their derivatives (25.28%), 265 phenolic acids (17.73%), 168 lipids (11.24%), 114 flavonoids (7.63%), 103 organic acids (6.89%), 92 alkaloids (6.15%), 80 nucleotides and their derivatives (5.35%), 55 lignans and coumarins (3.68%), and 42 terpenoids (2.81%), 23 steroid (1.54%) and 75 metabolites belonging to other categories (11.71%) (Fig. 5A). Notably, the flavonoid category was further subdivided into nine chalcones, 17 dihydroflavonoids, eight dihydroflavonols, 36 flavonoids, 40 flavonols, and four flavanols.

Figure 5 Metabolome analysis results.

(A) Composition of metabolite in Dai-Bai-Jie. (B) PCA score plots for all samples. (C) Heat map of DAMs in four groups of samples. (D) Venn diagram of DAMs across groups.

PCA was employed to illuminate the overall metabolite differences among the different groups. The results showed that principal component 1 (PC1, 38.39%) and principal component (PC2, 23.73%) accounted for 62.12% of the variance in the metabolic profile, indicating significant differences across four groups. The three samples within each group demonstrated high aggregation and good repeatability (Fig. 5B).

A total of 943 differential metabolites (DAMs) were detected using FC ≥ 2 or ≤ 0.5 and VIP > 1 as screening conditions, including 255 amino acids and their derivatives, 174 phenolic acids, 45 nucleotides and their derivatives, 79 flavonoids, 42 lignans and coumarins, 64 alkaloids, 30 terpenoids, 44 organic acids, 20 steroids and 83 lipids. Among them, there were one common DAMs shared of CR1, CR2, CR3, and CR4. Specifically, there were five unique DAMs in the comparison of CR1 vs. CR2, 273 unique DAMs in CR1 vs. CR3, 172 unique DAMs in CR2 vs. CR3, and 46 unique DAMs in CR3 vs. CR4 (Fig. 5D).

Figure 6 The volcano diagram and the k-means diagram of Metabolites.

(A) Volcano diagram of DAMs (CR1 vs. CR2). (B) Volcano diagram of DAMs (CR1vs. CR3). (C) Volcano diagram of DAMs (CR2 vs. CR3). (D) Volcano diagram of DAMs (CR3 vs. CR4). (E) The K-means analysis of all metabolites. The black line in the figure represents the average pattern of all metabolites in each class, and different colors represent different trend.

In the comparison of CR1 vs. CR2, a total of 627 DAMs were detected, of which 183 were down-regulated and 444 were up-regulated. Compared to CR1, the metabolite that significantly decreased in CR2 was gofruside, whereas the metabolite that significantly increased was 4-O-(2″-O-acetyl-6″-P-coumaroyl-β-D-glucopyranosyl)-P-coumaric acid (Fig. 6A). The metabolite protocatechuic acid 4-O-(2″-O-Vanilloyl) glucoside significantly decreased in CR3 compared to CR1, while eugenol significantly increased (Fig. 6B). A total of 449 DAMs were detected in CR2 vs CR3, with 377 down-regulated and 72 up-regulated. The metabolite 6,7-dimethoxy-2-[2-(4′-hydroxy-3′-methoxyphenyl)ethyl]chromone was significantly reduced in CR3 relative to CR2, while sinapine was significantly increased (Fig. 6C). Lastly, a total of 259 DAMs were found in the comparison between CR3 vs CR4, with 117 down-regulated and 142 up-regulated. The metabolite that showed a significant decrease in CR4 was rutin, while exhibited a significant increase when compared to CR3 (Fig. 6D). Cluster analysis was performed on the DAMs across the four groups. The differences among the four groups were pronounced; specifically, the phenolic acids were commonly more abundant in CR2, and flavonoids were commonly higher in the CR1 and CR2 compared to in the other groups. Additionally, the levels of amino acids and their derivatives were higher at CR3, while the contents of terpenes, nucleotides and their derivatives were higher in CR4 (Fig. 5C).

To gain a deeper understanding of the accumulation patterns of metabolites in Dai-Bai-Jie across different planting ages and altitudes, we employed k-means cluster analysis to categorize all the metabolites. The analysis revealed that the metabolites clustered into six distinct groups (Fig. 6E). Notably, classes 1 and 6 exhibited the highest concentration of metabolites in CR2, with class 6 containing the largest number of metabolites among all six classes. Classes 2 and 4, on the other hand, demonstrated the highest abundance of metabolites in CR3. Class 3 was characterized by the highest amount of metabolites in CR4, while class 5 displayed the highest concentration of metabolites in CR1. This categorization provides valuable insights into the specific patterns of metabolite accumulation within each growth year and altitude, enabling us to further investigate their potential biological significance.

Comparative metabolomic analysis aiming to flavonoids and flavonoid biosynthesis-related genes among the different plantation age and locality

A total of 114 flavonoids were detected from Dai-Bai-Jie, including 34.21% flavonols, 31.58% flavonoids, 14.91% dihydroflavonoids, 7.02% dihydroflavonols, 7.89% chalcone, 3.50% flavanols, 0.88% flavonols, of which 79 flavonoids were differentially accumulated. Based on K-means analysis, nine flavonoids, including 3′,5-Dihydroxy-4′,6,7-trimethoxyflavanone, acacitin-7-O-galactide, robiniin-7-O-galactoside, phelamurin, huangbaioside, eriodictyol-7-O-glucoside, exhibited a relatively high accumulation in class 2 for CR2. 15 flavonoids including 3′, 4′, 7-trihydroxyflavone, cirsimaritin, hesperetin-7-O-glucoside, quercetin, exhibited a relatively high accumulation in class 6 for CR2. Six flavonoids including kaempferol-7-O-glucuronid, hesperetin-7-O-(6″-malonyl) glucoside, quercetin-3-O-(6″-O-galloyl) galactoside, myricetin-3-O-rhamnoside (Myricitrin), diosmetin-7-O-glucuronide, syringetin-7-O-glucoside, exhibited a relatively high accumulation in class 2 for CR3. Ten flavonoids including Rutin, hesperetin-5-O-glucoside, isorhamnetin-3-O-rhamnoside, quercetin-3-O-robinobioside, exhibited a relatively high accumulation in class 4 for CR3. Five flavonoids including 3-Hydroxy-4′,5,7-trimethoxyflavanone, aromadendrin-7-O-glucoside, eriodictyol-8-C-glucoside, dihydromyricetin-3-O-glucoside, taxifolin-3′-O-glucoside, exhibited a relatively high accumulation in class 3 for CR4. 34 flavonoids including rhamnazin, quercetin-3,4′-dimethyl ether, limocitrin-7-O-glucoside, kumatakenin, exhibited relatively high accumulation in class 5 for CR1.

To gain a deeper understanding of the molecular mechanisms underlying the differential accumulation of flavonoids across various planting year and planting environments, we conducted a comprehensive analysis of the expression patterns of genes involved in flavonoid metabolism. KEGG analysis revealed that the 15 flavonoids exhibiting differential accumulation were mapped to multiple biosynthetic pathways, including the flavonoid biosynthesis pathway (KO00941), flavonol biosynthesis pathway (KO00944), as well as the broader metabolic pathway (KO01100) and secondary metabolite biosynthesis pathway (KO01110) (Fig. 7A).

Figure 7 Flavonoid synthesis pathway and network diagram.

(A) Flavonoid synthesis pathway. (B) Network diagram of flavonoids and differential genes.

Correlation analysis was conducted between DAMs mapped to the KEGG pathway and the corresponding DEGs on the pathway, and the correlation >0.8 or <−0.8 and the P-value < 0.05 as the screening conditions. The analysis revealed complex regulatory relationship among phenylalanine ammonia-lyase (PAL Cluster-63886.0, Cluster-63886.1), 4-Coumarate: Coenzyme A Ligase (4CL, Cluster-58688.4, Cluster-62808.3), lavonol synthase (flavanol synthase (FLS), Cluster-46899.18, Cluster-46899.5, Cluster-50957.2, Cluster-57391.0, C12RT1(Cluster-45854.0)and the metabolites hyperin, lonicerin, vicenin-2, nicotiflorin, querceti, luteolin-7-O-(6″-malonyl) glucoside, and hesperetin-7-O-glucoside (Fig. 7B).

Taxonomic features of the rhizosphere microbes of Dai-Bai-Jie

Plants recruit specific root-associated microbes that enable them to deliver photosynthates and root exudates to their root microbiome, thereby stimulating plant growth and productivity (Lareen, Burton & Schäfer, 2016). Studies have indicated that the composition of microbial communities at roots, the so-called root microbiome, can have significant impacts both on plant development and their stress tolerance (Mendes et al., 2011; Panke-Buisse et al., 2015).

The coverage index between the bacterial and fungal sample groups exceeded 0.965, indicating that the sequencing was representative and accurately reflected the bacterial and fungal diversity of the samples. The four groups of rhizosphere soil bacteria involved a total of 40 phyla, 71 classes, 154 orders, 300 families, and 695 genera, and fungi comprised 13 phyla, 61 classes, 168 orders, 406 families, and 875 genera. The dominant bacterial phyla in the rhizosphere soils included Crenarchaeota, Acidobacteriota, Chloroflexi, Firmicutes, Proteobacteria were the dominant bacteria in the rhizosphere soils, whereas the predominant fungal phyla were Ascomycota, Basidiomycota, Mortierellomycota, Glomeromycota, Chytridiomycota, and Rozellomycota.

We investigated the richness indices (alpha diversity, ACE, Chao1) and the Shannon diversity index of the microbial community, as well as the number of operational taxonomic units (OTUs) across all samples. There were no significant differences in the Shannon, Chao1, and ACE indices of rhizosphere microorganisms among the four groups (Table 2).

Table 2 Diversity index of microbial communities in roots soils.

Sample	Shannon	Chao1	ACE	Goods_coverage	
16s	CM1	10.275 ± 0.133	4,844.719 ± 755.638	4,912.204 ± 744.152	0.972 ± 0.006	
CM2	9.546 ± 0.128	4,449.788 ± 173.103	4,563.577 ± 259.590	0.972 ± 0.002	
CM3	9.507 ± 0.159	4,580.545 ± 122.909	4,696.261 ± 115.363	0.972 ± 0.002	
CM4	10.039 ± 0.067	4,857.131 ± 150.655	4,936.003 ± 163.463	0.971 ± 0.002	
ITS	CM1	5.894 ± 0.317	1,207.431 ± 60.164	1,227.965 ± 58.934	0.997 ± 0.001	
CM2	5.267 ± 0.459	1,152.941 ± 259.092	1,185.874 ± 266.003	0.997 ± 0.001	
CM3	5.722 ± 0.276	1,408.72 ± 171.792	1,445.749 ± 164.948	0.996 ± 0.001	
CM4	6.320 ± 0.133	1,424.764 ± 70.520	1,449.286 ± 80.612	0.996 ± 0.001	

A total of 1,952 bacterial operational taxonomic units (OTUs) and 5,230 fungi were detected in the rhizosphere microbiome. The co-possessed bacteria in the four rhizosphere soils are 2986 OTUs, 721 are unique to CM1, 406 are unique to CM2, 497 are unique to CM3, and 620 are unique to CM4 (Fig. 8A). The co-possessed fungi in the four rhizosphere soils are 5677 OTUs, 383 are unique to CM1, 223 are unique to CM2, 263 are unique to CM3, and 406 are unique to CM4 (Fig. 8B).

Figure 8 Venn diagram and the relative abundance of phylum and genus among CM1, CM2, CM3, and CM4 in rhizosphere soil of Dai-Bai-Jie.

(A) Venn diagram of bacterial. (B) Venn diagram of fungus. (C) relative abundance of bacterial phylum. (D) Relative abundance of fungal phylum. (E) Relative abundance of bacterial genus. (F) Relative abundance of fungal genus.

Community composition analysis revealed that the compositions were similar among all twelve rhizosphere soils samples at the phylum level. Excluding CM3.3, the abundance of Acidobacteriota in CM2 and CM3 was significantly higher than in CM1 and CM4 (Figs. 8C, 8D). However, the community compositions presented some differences among all twelve rhizosphere soils at the genus level (Figs. 8E, 8F).

Discussion

The growth duration is the most critical factor affecting the quality of medicinal plants. Until now, the harvesting period of Dai-Bai-Jie has primarily centered on biomass accumulation, with the accumulation of bioactive components remaining unknown. Despite numerous research reports have examined the metabolites and anti-tumor properties of G. tenacissima, the majority of these studies have not specifically targeted Dai-Bai-Jie, largely due to inaccuracies in plant identification (Li et al., 2014; Li et al., 2023). Up to now, little is known about the chemical composition and active ingredients of Dai-Bai-Jie (Liao et al., 2016; Pang et al., 2018; Li et al., 2017). This highlights the necessity for further scientific investigation to comprehensively understand the growth patterns and accumulation of bioactive components in Dai-Bai-Jie.

Plant-wide target detection was conducted using the high-resolution mass spectrometer AB Sciex TripleTOF6600 for qualitative analysis of mixed samples. Subsequently, quantitative analysis was performed using the AB Sciex 6500 QTRAP. This approach combines the advantages of both non-targeted and targeted metabolomics, employing high-resolution mass spectrometry for accurate qualitative detection and utilizing a triple quadrupole mass spectrometer with high sensitivity, specificity, and excellent quantitative capabilities as a supplementary tool. In this study, a comprehensive metabolic profiling of Dai-Bai-Jie was conducted using UPLC-MS/MS widely-targeted metabolomics analysis. A total of 1,495 metabolites were successfully identified, signifying the rich metabolite content of Dai-Bai-Jie. These metabolites are likely to form the pharmacological material basis for the medicinal properties of Dai-Bai-Jie. Additionally, 943 DAMs were detected from four group samples obtained from distinct locations and three different planting age, which suggests quality variations among them.

Flavonoids and total polyphenols were major contributors for detoxification of Dai-Bai-Jie (Zhang et al., 2023a; Zhang et al., 2023b). We detected a diverse array of secondary metabolites, including flavonoids, phenolic acids, alkaloids, and terpenoids, which may potentially contribute to its antioxidant and anti-inflammatory activities.

When comparing the accumulation of metabolites across different planting ages, it was observed that the total metabolite content in CR2 and CR3 was relatively abundant. Additionally, flavonoid levels were generally higher in CR1 and CR2. To achieve an optimal balance between biomass, economic benefits, and the biological activity of Dai-Bai-Jie, it is recommended that two-year harvesting serves as the optimal strategy.

Despite originating from the same planting age, samples CR3 and CR4 exhibited inconsistent trends in metabolite accumulation, revealing a total of 259 DAMs. This variation can be attributed to diverse environmental factors, including altitude, temperature, and soil conditions. Although the number of DAMs identified was lower compared to those observed between different years, it nonetheless underscores the significant impact of the environment on the accumulation of secondary metabolites in Dai-Bai-Jie. Furthermore, it suggests that cultivation at lower altitudes may result in a diminished abundance of secondary metabolites. This could be due to the influence of lower temperatures at higher altitudes, which may induce the expression of resistance genes, thereby promoting the accumulation of secondary metabolites. Consequently, in the future large-scale introduction and cultivation of Dai-Bai-Jie, high-altitude conditions should be carefully considered.

Based on plant-wide target metabolome analysis, flavonoids are identified as the predominant secondary metabolites in Dai-Bai-Jie. Notably, the flavonoid content is significantly greater in plants cultivated for two and three years compared to those cultivated for one year. This finding is generally consistent with the flavonoid accumulation patterns observed in most medicinal plants (Kuang, Jiang & Wang, 2020; Yuan et al., 2022).

Numerous flavonoids isolated from Dai-Bai-Jie have exhibited significant biological activities. Specifically, hesperetin-7-O-glucoside has been demonstrated to effectively modulate the gut microbiota composition and bile acid metabolism in murine models (Wu et al., 2022). The antioxidative, antihypertensive, antidiabetic, anti-inflammatory, and cardioprotective activities of rutin were reported, while rutin pretreatment before administration of ethanol can afford significant protection against mucosal hyperemia, necrosis, edema, and mucosal or submucosal hemorrhage (Akash et al., 2024; Chua, 2013; Nicola et al., 2024). Quercetin is known to possess both mast cell stabilizing and gastrointestinal cytoprotective activity (Anand David, Arulmoli & Parasuraman, 2016; Catalina et al., 2016).

The flavonoid content in Dai-Bai-Jie varies significantly with its plantation age, which may be the result of DEGs patterns of genes involved in flavonoid biosynthesis. To date, flavonoid biosynthetic pathway has been extensively studied, with the genes encoding enzymes involved in this pathway and their respective functions having been verified in numerous plants. Flavonoids, flavonols, and lignin are synthesized through various branching pathways originating from the phenylpropane biosynthetic pathway (Froemel et al., 1985). We screened nine DEGs related to flavonoid biosynthesis from Dai-Bai-Jie, PAL, 4CL, FLS, and C12RT1 including. PAL catalyzes the first step in the phenylpropanoid pathway and plays an important role in the biosynthesis of phenylpropanoid and flavonoid compounds (Levy, Sarkissian & Scriver, 2018; Oyanagi & Ozeki, 2001). 4CL is the last enzyme in the general biosynthetic pathway of phenylpropane compounds, which catalyzes cinnamic acid and its hydroxyl or methoxy derivatives to generate corresponding coenzyme A esters (Cao et al., 2023; Lavhale, Kalunke & Giri, 2018). These intermediate products then enter the biosynthetic pathway of phenylpropane derivatives (Tian et al., 2017). Flavanol synthase (FLS) is a key enzyme specific to the flavonol pathway, which converts dihydroflavonol into the corresponding flavonol by introducing a double bond between C-2 and C-3 of the C-ring (Forkmann et al., 1986; Shi et al., 2021).

Correlation analysis conducted on flavonoid DAMs mapped to the KEGG pathway revealed that the expression patterns of the genes PAL, 4CL, and FLS exhibited a consistent trend with the accumulation of nicotiflorin and lonicerin. Similarly, hesperetin-7-O-glucoside displayed a comparable trend with C12RT1. These DEGs may serve as key regulators of the distinct accumulation patterns of flavonoid metabolites in Dai-Bai-Jie.

The RT-qPCR results showed that the expression trend of the key enzyme genes in the biosynthetic pathway of flavonoids in Dai-Bai-Jie was consistent with the results of transcriptome sequencing, thereby confirming the reliability of the transcriptome data.

In general, the age of plantation has been shown to induce changes in soil nutrient content and pH, subsequently affecting the composition and diversity of soil bacterial and fungal communities. For instance, Na et al. (2016) reported that fungal diversity decreased with the cultivation going on from 5 a to 10 a of Lycium bararum L. whereas bacterial diversity remained relatively unchanged. Conversely, Li & Xu (2020) observed a significant increase in bacterial diversity and a decrease in fungal diversity in lily soil with increasing planting years. However, in our study on Dai-Bai-Jie, we did not detect any significant differences in the Shannon, Chao1, or ACE indices of rhizosphere microorganisms across different plantation ages and localities. This inconsistency suggests that the underlying mechanisms governing microbial community dynamics in the rhizospheres of Dai-Bai-Jie may differ from those observed in other plant species, possibly due to the relatively short introduction period of Dai-Bai-Jie.

The absence of significant changes in microbial diversity warrants further investigation, particularly from the perspectives of soil nutrients, pH, and moisture content.

In summary, this study comprehensively characterized the disparities in flavonoid metabolite profiles and abundances across varying cultivation environments and plantation age through integrated transcriptome and metabolome analyses. Key genes intricately associated with the differential accumulation of flavonoids were identified. The results laid a foundation for further regulation of the effective components and support the formulation of scientifically harvesting practices for Dai-Bai-Jie.

Conclusions

In summary, this study thoroughly characterised the disparities in metabolites and flavonoid metabolite profiles and abundances across varying cultivation environments and plantation ages through integrated transcriptome and metabolome analyses. A total of 1,495 metabolites were identified using UPLC-MS/MS from Dai-Bai-Jie across three different planting durations (one year, two years, and three years) at two distinct localities. Among these, 943 DAMs were detected. A total of 114 flavonoids were identified, of which 79 exhibited differential accumulation. The total metabolite content in CR2 and CR3 was relatively abundant, and flavonoid levels were generally higher in CR2 and CR3. Therefore, it is recommended that harvesting at two years of age be considered the optimal strategy. Key genes intricately associated with the differential accumulation of flavonoids were identified. We found a complex regulatory relationship among phenylalanine ammonia-lyase (PAL; Cluster-63886.0, Cluster-63886.1), 4-Coumarate: Coenzyme A Ligase (4CL; Cluster-58688.4, Cluster-62808.3), flavonol synthase (FLS; Cluster-46899.18, Cluster-46899.5, Cluster-50957.2, Cluster-57391.0, C12RT1; Cluster-45854.0), and the metabolites hyperin, lonicerin, vicenin-2, nicotiflorin, quercetin, luteolin-7-O-(6″-malonyl) glucoside, and hesperetin-7-O-glucoside. Different planting ages and localities did not result in significant differences in the Shannon, Chao1, or ACE indices of the rhizosphere microorganisms associated with Dai-Bai-Jie. The results establish a foundation for further regulation of pharmacological components and provide support for the development of scientific harvesting practices for Dai-Bai-Jie.

Supplemental Information

Supplemental Information 1 qRT-PCR data

Supplemental Information 2 Metabolome of Dai-bai-jie

Supplemental Information 3 MIQE Checklist

Supplemental Information 4 Diversity index of microbial communities in roots soils

We acknowledge South Medicine Garden of Xishuangbanna Dai Autonomous Prefecture for providing the material collection site.

Additional Information and Declarations

Competing Interests

Author Contributions

Data Availability

The authors declare there are no competing interests.

Mengqi Wang conceived and designed the experiments, performed the experiments, analyzed the data, authored or reviewed drafts of the article, and approved the final draft.

Yunxia Gu performed the experiments, prepared figures and/or tables, and approved the final draft.

Liming Shan performed the experiments, analyzed the data, prepared figures and/or tables, and approved the final draft.

Chunyu Li analyzed the data, prepared figures and/or tables, and approved the final draft.

Ertai Yuan analyzed the data, prepared figures and/or tables, and approved the final draft.

Ge Li conceived and designed the experiments, authored or reviewed drafts of the article, and approved the final draft.

Xiaoli Liu conceived and designed the experiments, authored or reviewed drafts of the article, and approved the final draft.

The following information was supplied regarding data availability:

The data is available in the Supplemental Files.

The data is available at Zenodo: Wang, M., Gu, Y., Shan, L., Li, C., Yuan, E., Li, G., & Liu, X. (2025). Gongronemopsis tenacissima (Dai-Bai-Jie) raw metabonomics data for different planting years and locations (Data set). Zenodo. https://doi.org/10.5281/zenodo.17222367

The raw sequence data are available at NCBI BioProject: PRJNA996325.

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
