# Peer review of "Multiomics analysis reveals flavonoid accumulation and biosynthesis across different cultivation years and localities of Gongronemopsis tenacissima (Dai-Bai-Jie)"

_PeerJ, doi:10.7717/peerj.20439_

## Round 0.1 · original submission · Major Revisions

· Academic Editor

Major Revisions

**Language Note:** The review process has identified that the English language must be improved. PeerJ can provide language editing services - please contact us at [email protected] for pricing (be sure to provide your manuscript number and title). Alternatively, you should make your own arrangements to improve the language quality and provide details in your response letter. – PeerJ Staff

Reviewer 1 ·

Basic reporting

Author's have done a great job however there are few comments in the addition comments sections which needs to be addressed

Experimental design

Experiment is fine and within the scope of journal purely

Validity of the findings

No comments

Additional comments

This study systematically investigated the accumulation patterns and regulatory mechanisms of flavonoids in Dai-Bai-Jie(Gongronemopsis tenacissima) under different planting years and geographical environments using a multi-omics approach. The research results provide a scientific basis for optimizing the cultivation and harvesting strategies of Dai-Bai-Jie and lay the foundation for further development of its medicinal value. Overall, The research design of this paper is well-structured, the data is comprehensive, and it demonstrates a certain level of innovation and practical significance. However, there are some areas where improvements are needed,I would suggest a major revision.

1.The manuscript does not fully address the implications of the findings beyond the scientific domain, mainly reflected in the abstract and the discussion section.

2. The discussion part is not in-depth enough, which is manifested as follows: comparison with other similar studies should be supplemented, especially regarding the similarities and differences in the accumulation patterns and regulatory mechanisms of flavonoids in different plant species.

The possible mechanisms for the reduction of metabolite accumulation in low-altitude areas.

3. In terms of language, some sentences have incorrect structures, such as line 422-426

4. the formats of some references are inconsistent, such as the abbreviations of journal names, the format of the DOI, and so on.

5. Perform thorough language editing for clarity and conciseness.

·

Basic reporting

The English is understandable in most places, but sometimes not very clear. In many places, a certain sloppyness can be observed, be it in the literature citations with arbitrary use of majuscules and minuscules, or use of the old name “Marsdenia tenacissima” instead of “Gongronemopsis tenacissima”, Italics are not applied consistently for genera and species names, etc.

The Introduction is mostly understandable for a person not working in the metabolomics field. The papers cited are relevant, but I don’t know the literature well enough to decide whether it is complete. Explanation of the procedures used is difficult for someone not working directly in the field.

In general, the structure conforms to PeerJ standard, but not all abbreviations are explained at first use.
Subheadings are not bold, and not followed by a period.

There is no Background section.

There are no Acknowledgements.

Figures, Tables, and Raw Data are fine (as far as I can judge).

Experimental design

Research is original and in the scope of PeerJ.
The research question is defined, and relevant to people using the drug and those wanting to protect the species in the wild. Whether it is a meaningful contribution to the wider field of drug research I cannot judge.

The investigation is carried out almost with an overkill of modern methods, certainly with a high technical standard – there is no relevant ethical question involved.

Sometimes the description of methods reads like a lab protocol rather than a text and is thus probably suitable for replication.

Validity of the findings

Underlying data have been provided.

Conclusions are valid, and discussed within the framework of the paper. However, they could be formulated more convincingly.

---

## Round 0.2 · Minor Revisions

· Academic Editor

Minor Revisions

**Language Note:** When you prepare your next revision, please either (i) have a colleague who is proficient in English and familiar with the subject matter review your manuscript, or (ii) contact a professional editing service to review your manuscript. PeerJ can provide language editing services - you can contact us at [email protected] for pricing (be sure to provide your manuscript number and title). – PeerJ Staff

Reviewer 1 ·

Basic reporting

I think authors have addressed my comment carefully; therefore, I have no further comments.

Experimental design

-

Validity of the findings

-

·

Basic reporting

The English is better, but the paper is so sloppily formatted that it is annoying. Unfortunately, it is beyond my judgment whether the research has been conducted in the same style.

The use of Gongronemopsis as a suitable genus should be from the beginning, and the reference to Marsdenia should only be made at first mention of the plant, or, if a particular literature is discussed, it can be ("under Marsdenia....).

The Subheadings (3.1, 3.2 ........) are not bold, and not followed by a period.

Experimental design

-

Validity of the findings

-

---

## Round 0.3 · Minor Revisions

· Academic Editor

Minor Revisions

Authors have addressed most of the comments. However, there are still minor corrections that need to be done meticulously. Check the annotated file for corrections.

·

Basic reporting

As this is the third round of review, I don't write about generalities again. The authors have corrected most, though not all, points criticized in the last two rounds. Some corrections have been written directly onto the manuscript.

Experimental design

See the first review

Validity of the findings

See the first review

---

## Round 0.4 · Minor Revisions

· Academic Editor

Minor Revisions

There still remain some formatting issues (e.g. double spaces or unintended hard spaces).

In addition, the Section Editor, has commented and said:

"1) The identification of metabolites based on LC-MSn is not described in sufficient detail to be reproducible.

2) The authors should deposit raw data and used resources for the identification of metabolites (such as compound lists, R scripts) on a public repository such as Zenodo."

Please address these points in your next revision.

---

## Round 0.5 · accepted · Accept

· Academic Editor

Accept

The reviewers are generally satisfied, but please note the remaining editorial comments in the manuscript.

·

Basic reporting

the English is now improved, the other points have been okay before.

Experimental design

not changed, fine

Validity of the findings

not changed, fine

Additional comments

see comments on the ms